# Enhancing Network Intrusion Detection Using an Ensemble Voting Classifier for Internet of Things

**DOI:** 10.3390/s24010127

**Published:** 2023-12-26

**Authors:** Ashfaq Hussain Farooqi, Shahzaib Akhtar, Hameedur Rahman, Touseef Sadiq, Waseem Abbass

**Affiliations:** 1Faculty of Computing and AI, Air University, Islamabad 44000, Pakistan; ashfaq.hussain@mail.au.edu.pk (A.H.F.); 220308@students.au.edu.pk (S.A.);; 2Centre for Artificial Intelligence Research, Department of Information and Communication Technology, University of Agder, 4879 Grimstad, Norway; 3Department of Electrical Engineering and Computer Engineering, Capital University of Science and Technology, Islamabad 44000, Pakistan

**Keywords:** machine learning (ML), synthetic minority over-sampling technique (SMOTE), network intrusion detection system (NIDS)

## Abstract

In the context of 6G technology, the Internet of Everything aims to create a vast network that connects both humans and devices across multiple dimensions. The integration of smart healthcare, agriculture, transportation, and homes is incredibly appealing, as it allows people to effortlessly control their environment through touch or voice commands. Consequently, with the increase in Internet connectivity, the security risk also rises. However, the future is centered on a six-fold increase in connectivity, necessitating the development of stronger security measures to handle the rapidly expanding concept of IoT-enabled metaverse connections. Various types of attacks, often orchestrated using botnets, pose a threat to the performance of IoT-enabled networks. Detecting anomalies within these networks is crucial for safeguarding applications from potentially disastrous consequences. The voting classifier is a machine learning (ML) model known for its effectiveness as it capitalizes on the strengths of individual ML models and has the potential to improve overall predictive performance. In this research, we proposed a novel classification technique based on the DRX approach that combines the advantages of the *D*ecision tree, *R*andom forest, and *X*GBoost algorithms. This ensemble voting classifier significantly enhances the accuracy and precision of network intrusion detection systems. Our experiments were conducted using the NSL-KDD, UNSW-NB15, and CIC-IDS2017 datasets. The findings of our study show that the DRX-based technique works better than the others. It achieved a higher accuracy of 99.88% on the NSL-KDD dataset, 99.93% on the UNSW-NB15 dataset, and 99.98% on the CIC-IDS2017 dataset, outperforming the other methods. Additionally, there is a notable reduction in the false positive rates to 0.003, 0.001, and 0.00012 for the NSL-KDD, UNSW-NB15, and CIC-IDS2017 datasets.

## 1. Introduction

Computing and mobile devices have become an integral part of our daily lives, and our reliance on this technology is substantial. However, the future is moving toward the metaverse, driven by an insatiable desire for technology adoption. It is increasingly clear that future generations will inhabit a multiverse, a concept that extends beyond a single Internet-connected device like a mobile phone [1]. Instead, it represents the convergence of various technologies, including the Internet of Things (IoT), digital twins, blockchain, augmented reality (AR), virtual reality (VR), extended reality (XR), 5G/6G networks, cloud computing, high-performance computing, artificial intelligence, and machine learning (AI/ML). These technologies share a common goal: enabling a multiverse existence. While the current generation focuses on mobile devices, future technologies will integrate multiple sensors on both the human body and in the surrounding environment to blend virtual environments with reality [2]. Moreover, the future envisions connecting everything, empowering individuals to control their surroundings through gestures and inhabit multiple virtual and augmented worlds simultaneously. Recent advancements in 5G, cloud computing, edge computing, high-performance computing, blockchain, and AI hold the potential to turn the metaverse into a reality by integrating numerous IoT devices [3].

Smart cities encompass a range of advanced technologies, such as intelligent transportation, Industry 4.0, smart healthcare, smart homes, and smart banking, among others, which demand high levels of data security while also aiming to enhance citizens’ quality of life. IoT applications are playing a significant part in real-world scenarios by enabling autonomous operations and communication. This, in turn, is promoting and enhancing the use of various services in daily life. With the progress of information and communication technology (ICT) and the widespread adoption of sensor technology, IoT is being increasingly utilized in various domains, such as healthcare, smart cities, and intelligent power grids, among others, to efficiently manage resources and enable ubiquitous sensing. Current IoT systems are susceptible to various types of security attacks, mainly because devices can be accessed from anywhere via the Internet and the use of lower-level security measures. Attackers can manipulate and cause harm to critical infrastructures, such as essential sensors, moving vehicles, and nuclear facilities, which has heightened the security issue with the smart cities network compared to traditional networks. The primary challenge for existing technologies lies in addressing security concerns, as we anticipate a significant increase in the attack surface due to the adoption of alternative network paradigms.

The IoT-enabled metaverse has inherited vulnerabilities that are adopted from the IoT communications. Attackers can degrade the performance of the system by launching different types of attacks as shown in Figure 1. In this regard, extensive research efforts have been devoted to the exploration of intrusion detection systems (IDSs) aimed at identifying and thwarting malicious activities within computer networks [4,5,6,7]. Signature-based IDSs are based on predefined rules, either static or adaptable, to detect potential network attacks. However, adversaries employ sophisticated techniques to obscure their malicious intentions, rendering these systems ineffective in detecting zero-day attacks [8,9]. Zero-day attacks, being previously unknown and lacking discernible patterns, pose a significant challenge for signature-based IDSs. In contrast, anomaly-based IDSs have emerged as a more efficient approach to combat such attacks, leveraging ML algorithms to differentiate between normal network traffic and malicious anomalies.

Network IDSs are advised to identify anomalous activities within the network. Signature-based NIDSs excel at detecting known attacks swiftly, leading to minimal response times. However, they falter in handling unknown attacks, resulting in a relatively low rate of false positives [10]. Conversely, anomaly-based NIDSs demonstrate prowess in detecting previously unidentified attacks by scrutinizing and categorizing network patterns. They exhibit heightened resilience to fluctuations in network behavior but tend to yield a higher rate of false positives in such circumstances. Numerous researchers have proposed a diverse array of anomaly-based NIDSs, subjecting them to evaluation using metrics such as false positives and detection rates [11].

The utilization of ML-based classification algorithms has experienced a notable surge in their application for constructing anomaly-based NIDSs. Researchers have put forth diverse ML-based models, assessing their efficacy using several available network datasets [12]. These models undergo training and subsequent evaluation against a range of performance metrics [12,13]. Prominent examples of these datasets encompass NSL-KDD, CIC-IDS2017, KDD-CUP, and UNSW-NB15, among others [14,15]. The evaluation process entails the meticulous examination of critical metrics, including accuracy, recall, F1-score, false positive rate, and precision. In this investigation, we present our proposed methodologies, scrutinized rigorously, employing the NSL-KDD, UNSW-NB15, and CIC-IDS2017 dataset across these aforementioned metrics. Traditional works concerning classification models typically lean toward employing a solitary classifier, such as naive Bayes [16], decision tree [17], support vector machines (SVMs) [18], random forest [19], or alternative meta-classifier approaches. However, recent years have witnessed a paradigm shift toward the utilization of multiple classifiers, fostering advancement in the field of ML. This shift reflects a growing recognition of the potential benefits derived from leveraging the collective knowledge and discernment offered by an ensemble of classifiers [20,21]. Using a combination of different algorithms like decision trees and gradient boosting in a voting classifier can be effective because it leverages the strengths of each model and may lead to better overall predictive performance, especially when the models have different biases and strengths.

Ensemble classifiers have garnered significant applicability in real-life scenarios, including remote sensing applications like automated greenhouses and water dams [22], along with various other domains [23,24]. Employing multiple classifiers in such contexts has showcased substantial advancements compared to the usage of a singular classifier. Likewise, recent studies have delved into the utilization of ensemble voting classifier techniques for NIDSs [25,26], alongside other domains. In this research paper, we present a novel DRX-based NIDS technique, leveraging an ensemble voting classifier approach. Specifically, we amalgamate three prominent ML classification algorithms: decision tree ((DT), random forest ((RF), and XGBoost (XGB), called the DRX. The core contributions of our research work are delineated below:We introduce an ensemble classifier, employing the voting classifier technique, to enhance classification performance. This ensemble classifier combines a decision tree, random forest, and XGBoost.The performance of the proposed ensemble classifier is evaluated using the state-of-the-art datasets: NSL-KDD, UNSW-NB15, and CIC-IDS2017.Evaluation metrics, including false positive rate, precision, recall, F1-score, and accuracy, are utilized to assess the efficacy of the suggested approach in terms of its performance.Experimental results demonstrate that the proposed methodology achieves an improved precision and accuracy while maintaining an acceptable level of false positive rate.The results provide evidence of the efficacy of the proposed methodology in handling multi-class datasets.

The subsequent sections of this paper are structured as follows: Section 2 provides an extensive review of the relevant literature. In Section 3, we present a concise overview of our proposed research methodology. The experimental setup and performance evaluation metrics are outlined in Section 4, while Section 5 delves into the conducted experiments and their respective results. Section 6 provides discussion on the results and efficacy of the proposed scheme. Finally, in Section 7, we offer conclusive remarks on our findings, along with potential avenues for future research.

## 2. Literature Review

In the current era of globalization, the demand for network-assisted applications has skyrocketed. Hence, our daily lives are heavily reliant on networks, and the pervasive use of IoT devices and services has significantly transformed our routines [27]. To meet the needs of end-users, next-generation network (NGN) systems are poised to harness the full potential of available networks, ushering in new dimensions and an enhanced utilization of network infrastructures [28]. In numerous applications, the secure transmission of data over network applications is of paramount importance. Any loss or compromise of data can lead to severe repercussions for critical decision support systems, including surveillance, healthcare, smart homes, and tracking systems for animals or children. Consequently, the deployment of NIDSs becomes imperative to identify malicious behavior exhibited by adversary-launched nodes [29].

Ongoing endeavors are being made to fortify the security of IoT and wireless sensor networks (WSNs), and cloud-based environments have yielded a range of NIDS-based solutions [30]. Unfortunately, as technology continues to advance, the prevalence of malicious actions has grown significantly. Consequently, these systems are inherently vulnerable and necessitate the deployment of sophisticated security mechanisms capable of effectively detecting and mitigating malicious activities within the network. As elucidated by [31], security emerges as an essential prerequisite for cyber–physical systems (CPSs). CPSs amalgamate diverse network paradigms to cater to a wide array of applications, including intelligent transportation systems, cloud-assisted healthcare systems, and IoT [32]. Therefore, to ensure the integrity and resilience of these interconnected systems, robust security measures become indispensable. In response, the NIDS plays a crucial role in detecting and mitigating the impact of malicious nodes that disrupt network performance and exhibit abnormal behavior.

To mitigate the impact of these attacks on network performance, various anomaly detection methodologies leveraging ML have been put forth. However, the majority of these approaches primarily focus on utilizing a single classifier, with only a few introducing ensemble techniques. Researchers have proposed and assessed the efficacy of employing two or more ML algorithms through a voting classifier. These techniques were evaluated using the NSL-KDD, UNSW-NB15, and CIC-IDS2017 dataset [33,34,35]. For instance, in [36], Belouch et al. introduced an ML-based NIDS that utilized the RepTree algorithm for detection and employed a two-classifier technique for feature extraction. Their approach achieved a notable accuracy level of 89%. Similarly, Liu et al. proposed a single-classifier technique and evaluated various ML algorithms, finding that XGBoost achieved a higher accuracy of 97% [37]. In another study, Khan et al. introduced a deep learning (DL) anomaly-based technique that achieved an accuracy of 91.23% [38]. However, these schemes were not tested against false positive rates. It is worth noting that the highest accuracy among these approaches was 97%, indicating that 3% of the malicious traffic went undetected, posing potential harm to the system and resulting in abnormal outcomes. Furthermore, Lian et al. employed recursive feature elimination (RFE) in combination with the DT algorithm and achieved an impressive accuracy of 99.23% [39]. This significant performance improvement can be attributed to the feature elimination technique utilized. However, there remains a need to evaluate these approaches against false positive rates and employ datasets with a more extensive range of features.

The review of the literature reveals that the ensemble classifier technique, which integrates multiple classification algorithms, surpasses the performance of single-classifier techniques, as indicated in Table 1. Gao et al. introduced an ensemble classifier voting scheme incorporating RF, DT, and deep neural network (DNN) algorithms [40]. This proposed scheme achieved an impressive accuracy of 84.23%. Notably, this approach synergistically combines DL algorithms with traditional ML algorithms, leveraging a structured dataset for evaluation. In a similar vein, Choobar et al. applied DL, yielding a remarkable accuracy of 98.50% [41]. However, it is important to acknowledge that this DL approach has not been evaluated in terms of its time consumption metric, which can be significantly higher than that of traditional ML techniques. Moreover, DL approaches are better suited for processing unstructured big data [42]. According to research findings, DL algorithms may exhibit lower efficiency when applied to structured network architectures, while demonstrating more suitability for unstructured network environments such as the IoT. Moreover, Alhowaide et al. introduced a novel technique employing an ensemble voting classifier, which effectively combined seven ML algorithms [43]. This approach achieved an impressive accuracy of 98%. However, it is worth noting that this scheme may not be optimized, as it combines several ML techniques, thereby increasing computational complexity without a proportional gain in accuracy.

Kunal et al. introduced an NIDS that employs an ensemble of random forest (RF), k-nearest neighbors (KNNs), and regression trees (RTs) algorithms [44]. Their proposed approach attained an impressive accuracy of 99.7% and an extremely low false positive rate (FPR) of 0.003. In a similar vein, Otoum et al. proposed an alternative ensemble classifier technique utilizing random forest (RF), DBa SCAN, and restricted Boltzmann machines (RBMs) algorithms [46]. This technique achieved a remarkable accuracy of 99.40% and an FPR of 0.013. Conversely, Yousefnezhad et al. presented an ensemble classifier technique combining k-nearest neighbors (KNN) and support vector machine (SVM) classification algorithms [33]. Their proposed scheme was evaluated using metrics such as accuracy, precision, F1-score, and recall, achieving an outstanding accuracy of 99.80%. In line with these studies, we evaluated our proposed anomaly-based NIDS, based on DRX, using the same metrics, and showcased a superior performance.

Further, Chen et al. proposed a similarity-aware IDS called ADSIM and evaluated it against precision, accuracy, recall, and F1-score [47]. They tested the proposed scheme using the MAWILab and CIC-IDS2017 dataset and the technique achieved 86.50% accuracy. Furthermore, Karna et al. presented a filter-based selection technique and used an ensemble classifier that was composed of DT, RT, and ET algorithms [48]. This technique was tested for NSL-KDD and CIC-IDS2017 and achieved 99.51% accuracy. It is worth noting that there are techniques that achieved higher accuracy while applying a single classifier, such as Kumar et al., who presented a single-classifier technique that used a random forest classification algorithm after applying an optimized sine swarm algorithm for feature selection [54] on the UNSW-NB15 dataset. The proposed technique achieved 98.15% accuracy with feature selection and 95.68% accuracy without feature selection.

Authors tested their proposed ensemble classifier technique using NSL-KDD and UNSW-NB15 datasets in [51,52,53]. In [51], authors presented a weighted-voting-based ensemble technique that used RF and AdaBoost. This technique received 89.50% accuracy and 9.23 FPR. The FPR is comparatively at the higher side compared to the other techniques mentioned in Table 1. In [52], authors proposed technique was a gain ratio feature evaluator (GRFE) that used a random committee ensemble scheme. This approach achieved a better accuracy of 98.80% and 0.033 FPR. Furthermore, in [53], authors came up with an enhanced flower pollination algorithm that ensembles DT, RF, and SVM classifiers. They evaluated it, achieving a higher accuracy. In our study, the ensemble classifier technique proposed in [25] was the best technique, achieving 99.80% accuracy. The presented scheme was a feature fusion and staking mechanism that ensembles DT, RF, and FPA algorithms.

The literature review shows that authors have used various combinations of ML algorithms to provide the optimal solution. Most of the work tested the proposed ensemble classifier technique using multiple datasets such as NSL-KDD, UNSW-NB15, and CIC-IDS2017. The reason behind using multiple datasets is to justify the efficacy of the scheme in different environments and in the diverse nature of IoT networks. The performance metrics that are used by most of the researchers are accuracy, precision, recall, F1-score, and FPR. In this work, we proposed an ensemble classifier technique called DRX and tested it using state-of-the-art datasets that are also used by others in their research, such as NSL-KDD, UNSW-NB15, and CIC-IDS2017. And, we tested our proposed technique for accuracy, precision, recall, F1-score, and FPR as targeted by most of the researchers to prove the performance of their proposed NIDS.

## 3. Proposed Network Intrusion Detection Framework

In recent times, blockchain technology has gained significant popularity and has been adopted in various next-generation applications, making it an appealing option for stakeholders across diverse industries [55]. Blockchain technology offers trust-free and decentralized solutions by storing data in a distributed manner using online-distributed ledgers. This eliminates the need for a trusted intermediary, and untrusted individuals can connect and exchange data in a verifiable manner. In the IoT network, the blockchain allows fog nodes to transact without relying on a central cloud authority, which helps to overcome the single point of failure problem. Software-defined networking (SDN), on the other hand, enables the remote, adaptive, and dynamic management of network data [56]. SDN separates the data-forwarding decision execution from the centralized controller, which can help with faster response times for attack detection in IoT security.

This paper presents a solution to the security attack detection problem in the IoT ecosystem by utilizing SDN, fog and edge computing, blockchain, and ML technologies. The proposed architecture is decentralized and employs SDN-enabled switches for dynamic traffic flow management to detect and mitigate attacks as shown in Figure 2. Attack detection is performed at the fog layer, and attacks are mitigated at the edge layer, with the help of edge computing and ML models. The use of blockchain technology facilitates the sharing of data among all fog nodes and the cloud server, enabling regular updates of the attack detection model. The proposed architecture ensures early detection, reduced storage requirement, lower latency, and less resource wastage, thereby improving the accuracy of attack detection in the metaverse ecosystem.

The researchers have dedicated significant efforts toward refining the accuracy of the model for effectively detecting malicious nodes within the network. The proposed solution introduces a voting classification methodology that leverages ensemble ML algorithms to discern between normal and malicious nodes. A visual representation of the proposed model can be observed in Figure 3. Artificial intelligence (AI)-based techniques use ML classification techniques to determine the outliers. Here, it is called the XAI model. The proposed XAI model is a voting classifier ensemble technique that capitalizes on the strengths of decision tree, random forest, and XGBoost models. The proposed ML-based NIDS technique works in five steps (1) Data is gathered from IoT environment at edge node (2) Pre-processing is performed on the data (3) Features are extracted from the pre-processed data (4) Data is split into train and test datasets (5) Model is trained using train dataset and validated the model using test dataset. These are further explained below:

### 3.1. Data Gathering

The NSL-KDD, UNSW-NB15, and CIC-IDS2017 datasets are utilized to evaluate the performance of the proposed ensemble classifier technique, which represents the latest version of the dataset. These datasets are easily available and already used by researchers for testing their proposed techniques (mentioned in Literature Review section).

#### 3.1.1. Representationof Intrusion Scenarios

NSL-KDD Dataset: NSL-KDD is a network intrusion detection system that is particularly derived from the KDD Cup 99 dataset. It contains a wide range of network traffic data that have categorized instances of both common and unusual invasive activity. The dataset offers a thorough depiction of network-based assaults and spans a large variety of intrusion situations.UNSW-NB15 Dataset: This dataset, created by the University of New South Wales, is intended for use in network-based intrusion detection inside Internet of Things environments. It captures a wide range of threats and abnormalities pertinent to contemporary IoT environments, including both legitimate and malicious network data. Given the present state of network security, the dataset’s emphasis on situations related to the Internet of Things is appropriate.CIC-IDS2017 Dataset: This dataset was developed by the Canadian Institute for Cybersecurity and is intended for use in network settings for intrusion detection. It includes actual network traffic, encompassing a wide range of assaults and routine operations. Because it represents modern network intrusion situations, the dataset is relevant and appropriate for assessing the suggested ensemble classifier in an authentic setting.

#### 3.1.2. Widespread Adoption and Availability of the Datasets

The scientific community has come to embrace and use these resources widely. These datasets are widely used by academics and industry professionals as industry standards for assessing intrusion detection system efficacy. This broad adoption improves the comparability of findings and makes it easier to comprehend the effectiveness of the suggested ensemble classifier in comparison to other approaches. Further, the scientific community may easily access these datasets, which encourages experiment transparency and reproducibility. Their uniform structure guarantees uniformity in assessment techniques, enabling an impartial and equitable appraisal of the suggested ensemble classifier.

### 3.2. Data Pre-Processing

Data pre-processing plays a crucial role in transforming raw data into a suitable format for effective utilization by ML techniques. To address computational complexity and enhance the performance of the IDS, feature selection techniques are commonly employed to eliminate irrelevant features [57]. In this study, the data undergo three key stages to ensure their suitability for classification techniques.

Firstly, the data undergo min–max normalization, scaling attribute values to a range between 0 and 1 to ensure uniformity across all attributes.Secondly, a label-encoding scheme is applied to convert string labels in the dataset into numeric values. For example, within the NSL-KDD dataset, the “normal” label is assigned a value of 0, while the “DoS”-labeled instances are assigned a value of 1, and so on.Lastly, the synthetic minority oversampling technique (SMOTE) is implemented [58,59] to address class imbalance. SMOTE oversamples the minority classes in the dataset by generating synthetic samples, resulting in a more balanced distribution of classes.

Following the completion of these pre-processing steps, the dataset is divided into an 80% training set and a 20% testing set. This combination of pre-processing techniques facilitates the generation of optimized training and testing datasets, which can be used effectively for training and evaluating the ensemble classifiers.

### 3.3. Machine Learning Algorithms

The proposed method incorporates an ensemble voting classifier that combines multiple ML algorithms. A decision tree, random forest, and XGBoost are ensembled in the proposed NIDS. Each of these algorithms offers unique strengths and capabilities, enhancing the overall performance of the ensemble classifier.

#### 3.3.1. Decision Tree

The decision tree algorithm is used by researchers as a stand-alone classifier. Mohammadi et al. introduced a feature selection technique and performed experiments using a decision tree and a multi-layer perceptron (MLP) for classification purposes [60]. Their findings indicated that the performance of the DT algorithm was superior compared to the MLP. In [17,60,61], a DT is used as a classification algorithm that utilizes attribute values to classify objects within a dataset. For instance, characteristics such as source and destination IP addresses, protocol type, and packet quantity can be utilized by the DT algorithm to determine whether the network traffic is normal or malicious.

The construction of the DT for an NIDS begins by selecting the most relevant features from the training data and utilizing them as the root node of the tree. Subsequently, branches are built based on the potential values of each feature, leading to subnodes representing the next feature to be evaluated. Eventually, the process reaches a leaf node that indicates the ultimate judgment of the tree based on the values of the characteristics.

Utilizing the acquired knowledge, the DT classification techniques employs a predefined set of conditions to categorize incoming network traffic as either legitimate or malicious. For instance, within the DT, a rule may state, “If the traffic is identified as malicious and the source IP address falls within a range of known malicious network IPs, it is classified as malicious”. When confronted with novel network traffic, the DT classification algorithm leverages these rules, in addition to others derived from the training data, to achieve precise and reliable classifications.

Consider the input sample denoted as X, representing the network traffic data, and the features Feature 1, Feature 2, …, Feature n within the sample, as well as the label of the class denoted as C (categorization into either normal or malicious). The DT algorithm operates as follows:

If the following equation is evaluated as true, the sample is classified as malicious:(1)(feature1=t1)and(feature2=u1)or(feature3=v1)and(feature4=w1)

Alternatively, if the specified condition is satisfied, the sample is classified as normal:(2)$$\left({\mathrm{feature}}_{5}=x_{1}\right)\;\mathrm{and}\;\left({\mathrm{feature}}_{6}=y_{1}\right)$$

If the output of the following condition is true, the class label C is determined to be malicious:  
(3)$$\left({\mathrm{feature}}_{7}=z_{1}\right)$$

By employing these rules, the DT algorithm effectively categorizes network traffic as either normal or malicious, enabling accurate intrusion detection.

#### 3.3.2. Random Forest

The random forest algorithm [62,63,64] is employed for the categorization of network traffic into either the normal or malicious class. Employing a particular ML technique, a combination of multiple DTs is utilized. This approach entails constructing a large number of DTs, where each tree is trained on a distinct random subset of the training data. The ultimate classification in the random forest is obtained by collectively aggregating the decisions made by all individual DTs, often through a majority voting mechanism.

To build a DT within the random forest, the algorithm starts by selecting the most relevant features from the training data and using them as the root node of the tree. Branches are then created based on the potential values of each feature, linking to subnodes that represent the subsequent features to be evaluated. This process continues until a leaf node is reached, which represents the final decision of the tree based on the values of the features.

The random forest algorithm generates multiple DTs using random subsets of the training data. When a new input sample is encountered, it is passed through each DT in the random forest for classification. The results from all the DTs are combined using majority voting.

In mathematical terms, the random forest algorithm can be represented as follows:(4)$$F=\mathrm{maj}\;\mathrm{voting}(F_{1},F_{2},\ldots ,F_{n})$$
where *F* represent the ultimate class label (normal or malicious), and *F*_1_, *F*_2_, …, *F*_n_ represent the class labels predicted by each individual DT.

To illustrate this concept, consider the following example: let us assume a random forest consisting of 100 DTs. When a new network traffic sample is evaluated, 80 DTs classify it as normal, while 20 DTs classify it as malicious. Through majority voting, the final classification of the random forest would be normal, indicating that the majority of the DTs agreed on the sample being normal.

#### 3.3.3. XGBoost

XGBoost (eXtreme Gradient Boosting) is a powerful ML technique used for classification and regression tasks  [62,65]. It is an ensemble learning method that combines the results of multiple weak models, typically DTs, to achieve a more accurate and robust prediction. XGBoost stands out for its utilization of gradient descent optimization, which effectively minimizes the loss function and enhances prediction accuracy.

In the context of NIDSs, XGBoost can be applied to classify network traffic as either legitimate or malicious. The first step involves creating a set of decision tree models using the XGBoost methodology. Each decision tree is trained on a subset of the training data. Through gradient descent optimization, the loss function is iteratively minimized to improve the overall model accuracy.

To classify a new input sample, the XGBoost model passes the sample through each DT. The predictions from these trees are then combined to obtain the final classification. This combination is achieved by taking a weighted sum of the individual DT predictions, with the weights assigned based on the optimization process. Each DT contributes to the final prediction, and the weight assigned to it reflects its relative importance and performance.

Mathematically, the XGBoost algorithm can be represented as follows:(5)$$C=\sum_{i=1}^{n}w_{i}C_{i}$$
where *C* represents the ultimate class label, which can be classified as normal or malicious. $C_{i}$ denotes the class label predicted by each specific DT, where *i* corresponds to the tree index. $w_{i}$ signifies the weight assigned to the individual DTs, which is determined during the optimization process. *n* signifies the total number of DTs within the XGBoost model. Through the process of passing a new input sample through each DT and combining their predictions using the respective weights, the XGBoost model generates a final prediction for the given sample. This approach harnesses the collective strength of multiple DTs, thereby enhancing the accuracy and dependability of the classification outcomes.

### 3.4. Proposed Ensemble Voting Classifier

An ensemble voting classifier combines predictions from multiple base models, often DTs, to obtain a more precise final forecast as described in Algorithm 1. Through a process known as majority voting, the predictions of the base models are aggregated to produce the ultimate projection. The ensemble classifier is trained using the same training data that are used for training the individual base models. Two primary categories of ensemble voting classifiers are hard voting and soft voting. In hard voting, the final outcome is determined by the class label most commonly predicted by the base models, such as normal or malicious. On the other hand, soft voting involves generating probability estimates as predictions from the base models. The class label with the highest average probability across all models is considered the final projection.

Mathematically, an ensemble voting classifier can be represented as follows.

Hard voting:  
(6)$$C=\mathrm{majority}\;\mathrm{voting}(C_{1},C_{2},\ldots ,C_{n})$$

Soft voting:  
(7)$$C=\mathrm{argmax}c\sum {i=1}^{n}P_{i,c}$$
where *C* signifies the ultimate class label predicted by the ensemble classifier. $C_{1},C_{2},\ldots ,C_{n}$ represent the class labels predicted by the individual base models within the ensemble classifier, employing hard voting. $P_{i,c}$ corresponds to the probability estimate for class label *c* predicted by the *i*th base model using soft voting. *n* denotes the total count of base models integrated within the ensemble classifier. Ensemble voting classifiers prove particularly advantageous when there is a requirement to consolidate predictions from multiple models, thereby enhancing the overall accuracy of the model. This approach proves valuable in situations where the individual base models exhibit high accuracy but may commit dissimilar types of errors. By amalgamating the predictions of the base models, the ensemble classifier effectively addresses these disparities, resulting in an improved overall performance.
**Algorithm 1** Ensemble voting classifier.  1:**Input:** Training data  $\mathrm{TD}=\left(a_{1},b_{1}\right),\left(a_{2},b_{2}\right),\ldots ,\left(a_{n},b_{n}\right)$, base model classifier type *T*, voting type *V*(hard or soft)  2:**Output:** Ensemble voting classifier EC  3:Initialize empty list EC  4:**for** (i=1) to *n* **do**  5:      Train-base model $M_{i}$ on D using classifier type *T*  6:      Append $M_{i}$ to EC  7:**end for**  8:**if** 
V=hard 
**then**  9:      **return** EC10:**else**11:      **return** Soft voting function with arguments EC12:**end if**

Where:The training data, denoted as TD, comprise pairs of input samples $a_{i}$ and their corresponding class labels $b_{i}$.The base model classifier, labeled as *T*, represents the specific type of classifier used, such as a DT or RF.The voting type, referred to as *V*, determines whether hard voting or soft voting is employed.The ensemble voting classifier, denoted as EC, takes the form of either a list of base models for hard voting or a function that executes soft voting for the case of soft voting.Each individual base model within the ensemble classifier is represented as $M_{i}$.

## 4. Performance Evaluation

### 4.1. Simulation Setup

The Jupyter notebook serves as the platform for simulating the proposed system. The solution is implemented using Python 3.9. The experiments are performed on a machine equipped with a fourth-generation Intel Core i5 processor and 8GB of RAM.

### 4.2. Performance Evaluation Metrics

Utilizing an ensemble classification approach, the proposed methodology incorporates three distinct ML algorithms: DT, RF, and XGBoost. This study employs a range of evaluation metrics for conducting experiments and analysis, encompassing the following:

Accuracy: Accuracy evaluates the proportion of correctly classified instances among all instances, providing an assessment of the classification scheme’s effectiveness in distinguishing malicious traffic. The accuracy is calculated using the following equation:(8)$$\mathrm{Accuracy}=\frac{\left(\mathrm{True}\;\mathrm{Negative})+(\mathrm{True}\;\mathrm{Positive}\right)}{\left(\mathrm{False}\;\mathrm{Positive})+(\mathrm{True}\;\mathrm{Negative})+(\mathrm{True}\;\mathrm{Positive})+(\mathrm{False}\;\mathrm{Negative}\right)}$$

Precision: Precision measures the ratio of correctly classified positive instances to the total instances classified as positive, indicating the classifier’s ability to identify specific classes accurately. The precision is calculated using the following equation:(9)$$\mathrm{Precision}=\frac{\left(\mathrm{True}\;\mathrm{Positive}\right)}{\left(\mathrm{False}\;\mathrm{Positive})+(\mathrm{True}\;\mathrm{Positive}\right)}$$

Recall: Recall, also known as sensitivity or true positive rate, quantifies the percentage of correctly identified positive instances out of all actual positive instances, reflecting the classifier’s ability to detect positive instances.
(10)$$\mathrm{Recall}=\frac{\left(\mathrm{True}\;\mathrm{Positive}\right)}{\left(\mathrm{True}\;\mathrm{Positive})+(\mathrm{False}\;\mathrm{Negative}\right)}$$

F1-Score: F1-score combines precision and recall into a single metric, serving as an evaluation measure for classification problems, particularly in imbalanced data scenarios.
(11)$$F1\text{-}\mathrm{score}=2\times \frac{\left(\mathrm{Precision})\times (\mathrm{Recall}\right)}{\left(\mathrm{Precision})+(\mathrm{Recall}\right)}$$

False Positive Rate (FPR): FPR measures the percentage of falsely classified negative instances out of all actual negative instances, providing insights into the rate of incorrect classifications of the negative class.
(12)$$\mathrm{FPR}=\frac{\left(\mathrm{False}\;\mathrm{Positive}\right)}{\left(\mathrm{False}\;\mathrm{Positive})+(\mathrm{True}\;\mathrm{Negative}\right)}$$

False Negative Rate (FNR): FNR calculates the ratio of incorrectly classified positive instances to all incorrect negative predictions, quantifying the rate of misclassifying positive instances as negative.
(13)$$\mathrm{FNR}=\frac{\left(\mathrm{False}\;\mathrm{Negative}\right)}{\left(\mathrm{True}\;\mathrm{Positive})+(\mathrm{False}\;\mathrm{Negative}\right)}$$

Average CPU Utilization Percentage:The average CPU utilization percentage is a metric used to quantify the average load on the CPU over a set of instances or observations.
(14)$$\mathrm{Avg}\_\mathrm{CPU}\_\mathrm{Utilization}=\frac{\sum_{i=1}^{n}\mathrm{CPU}\_{\mathrm{Utilization}}_{i}}{n}$$

Average Memory Utilization Percentage: The average memory utilization percentage is a metric used to express the average proportion of available memory consumed during a series of observations or tasks.
(15)$$\mathrm{Avg}\_\mathrm{Memory}\_\mathrm{Utilization}=\frac{\sum_{i=1}^{n}\mathrm{Memory}\_{\mathrm{Utilization}}_{i}}{n}$$

Average Training Time: The average training time is a performance metric that quantifies the average duration taken by a system or model to complete the training process across multiple instances.
(16)$$\mathrm{Avg}\_\mathrm{Training}\_\mathrm{Time}=\frac{\sum_{i=1}^{n}\mathrm{Training}\_{\mathrm{Time}}_{i}}{n}$$

## 5. Experiments and Results

The effectiveness of the proposed ensemble classifier based on DXR has been thoroughly evaluated using the state-of-the-art multi-class NSL-KDD dataset. This dataset is widely used for analyzing anomaly detection systems based on ML. The dataset is divided into an 80% training set and a 20% testing set. The proposed approach is assessed using well-established evaluation metrics, including accuracy, precision, recall, F1-score, and false positive rate (FPR), to measure the reliability and efficacy of the technique.

### 5.1. Results Based on NSL-KDD Dataset

Following the pre-processing stage, the NSL-KDD dataset is partitioned into distinct training and testing datasets. The distribution of samples across different classes is outlined in Table 2. The training dataset comprises 61,604 samples labeled as normal, 41,330 samples labeled as DoS, 10,300 samples labeled as probe, and 5506 samples labeled as R2L. Conversely, the testing dataset consists of 15,450 samples labeled as normal, 10,338 samples labeled as DoS, 2462 samples labeled as probe, and 1436 samples labeled as R2L.

Figure 4 presents the confusion matrix for the DRX classifier applied to the 80% training dataset of NSL-KDD. The results demonstrate that the proposed solution accurately classifies 61,443 samples as normal, 41,229 samples as DoS, 10,287 samples as probe, and 5301 samples as R2L. Similarly, Figure 5 displays the confusion matrix for the DRX classifier on the 20% testing dataset of NSL-KDD. The classifier effectively categorizes 15,329 samples as normal, 10,312 samples as DoS, 2438 samples as probe, and 1263 samples as R2L.

In Table 3, the performance results of the proposed anomaly-based NIDS are showcased for each class of the NSL-KDD dataset. The scheme demonstrates exceptional outcomes, boasting an accuracy of 99.69%, precision of 100%, recall of 100%, and F1-score of 100% for normal samples. Furthermore, it achieves remarkably high accuracy rates of 99.94%, 99.93%, and 99.97% for DoS, probe, and R2L samples, respectively.

The comprehensive evaluation of the proposed anomaly-based technique, specifically the DRX classifier, utilizing the NSL-KDD training dataset, reveals outstanding performance metrics. The proposed scheme achieves an impressive average accuracy of 99.88%, demonstrating its ability to accurately classify network traffic instances. Furthermore, it attains a remarkable precision rate of 99.25%, indicating its proficiency in correctly identifying relevant instances while minimizing false positives. The recall rate, measuring the ability to correctly detect instances of interest, reaches an excellent level of 99.0%. Finally, the F1-score, which combines precision and recall into a single measure, achieves an impressive score of 99.25%, highlighting the overall effectiveness of the proposed technique.

Table 4 presents a comprehensive analysis of the results obtained for each class of the NSL-KDD dataset after subjecting it to testing using the proposed DRX ensemble classifier. The proposed technique demonstrates exceptional performance with an accuracy rate of 98.93% for normal samples, showcasing its ability to accurately classify such instances. Notably, it achieves 100% precision, reflecting its capability to correctly identify relevant instances without any false positives. Additionally, the recall rate stands at 100%, indicating the classifier’s effectiveness in detecting all instances of interest within the dataset. The F1-score, which combines precision and recall, attains a perfect score of 100%, emphasizing the overall success of the proposed technique. For DoS, probe, and R2L samples, the proposed DRX classifier achieves accuracy rates of 99.83%, 99.71%, and 99.15%, respectively, showcasing its robustness in accurately classifying instances from these categories.

When considering the average scores achieved by the proposed DRX classifier across all classes, it demonstrates an impressive accuracy rate of 99.40%, showcasing its overall ability to classify network traffic instances effectively. The precision rate stands at 98%, indicating the classifier’s proficiency in correctly identifying relevant instances while minimizing false positives. The recall rate, measuring the ability to detect instances of interest, reaches an impressive level of 96.50%. Lastly, the F1-score, which combines precision and recall into a single measure, achieves a notable score of 98.50%, highlighting the overall effectiveness and reliability of the proposed DRX classifier when evaluated against the 20% test dataset of NSL-KDD.

### 5.2. Results Based on UNSW-NB15 Dataset

Initially, the UNSW-NB15 dataset is divided into 80% training and 20% testing datasets after pre-processing. The number of samples in each class is shown in Table 5. The number of samples acquired against normal, generic, exploits, fuzzers, DoS, reconnaissance, backdoor, shellcode, and worms labels is 74,387, 47,334, 35,445, 19,415, 13,061, 11,135, 1861, 1202, and 140, respectively, for training, while the number of samples obtained for testing is 18,613, 11,537, 9080, 4831, 3292, 2852, 468, 309, and 30, respectively.

Table 6 shows the results acquired after the training of the proposed anomaly-based NIDS for each class of the 80% UNSW-NB15 dataset. The proposed scheme achieves 100% accuracy for normal, generic, exploits, fuzzers, DoS, reconnaissance, and analysis samples, while it achieves 99.99% accuracy for backdoor and shellcode samples. The average score achieve by the proposed methodology is 99.94% accuracy, 99.92% precision, 99.92% recall, and 99.93% F1-score.

The results received from applying the proposed DRX ensemble classifier on the UNSW-NB15 20% test dataset are shown in Table 7. It shows that the proposed ML-based technique achieves 100% accuracy for normal, generic, exploits, fuzzers, DoS, and reconnaissance samples, while it achieves 99.99% accuracy for analysis and shellcode samples and 99.98% accuracy for backdoor samples.

The average score that is achieved by the proposed technique against accuracy, precision, recall, and F1-score is 99.99%, 99.98%, 99.99%, and 99.98%, respectively.

### 5.3. Results Based on CIC-IDS2017 Dataset

After pre-processing, the CIC-IDS 2017 dataset is divided into 80% training and 20% testing datasets. The number of samples after pre-processing in each class is shown in Table 8. The number of samples acquired against normal, DoS/DDoS, portscan, brute force, web attack, botnet, and infiltration labels is 91,607, 49,386, 18,491, 6223, 1558, 1694, and 29, respectively, for training, while the number of samples obtained for testing is 22,902, 12,346, 4623, 1556, 389, 424, and 7, respectively.

Table 9 depicts the results that are acquired for each class of the 80% CIC-IDS2017 training dataset after applying the proposed ensemble classifier. The results show that the proposed scheme achieves 100% accuracy for normal, web attack, brute force, infiltration, and portscan samples, while it accomplishes 99.99% accuracy for the DoS/DDoS and botnet ARES samples. The average score that is achieved by the proposed methodology for accuracy, precision, recall, and F1-score is 99.98%, 100%, 100%, and 100%, respectively.

The proposed ensemble classifier technique classifies 22,902, 389, 1556, 12,344, 7, 4620, and 423 normal, web attack, brute force, DoD/DDoS, infiltration, port scan, and botnet samples correctly, respectively.

Table 10 provides details about the results obtained for each class of the CIC-IDS2017 dataset after the testing of the proposed DRX ensemble classifier. The proposed technique attains 100% accuracy, precision, recall, and F1-score for normal, web attack, brute force, infiltration, and port scan samples, while it achieves 99.99% accuracy, precision, recall, and F1-score for DoD/DDoS and botnet samples.

### 5.4. Computational Efficiency of DRX Technique

Table 11 illustrates the DRX technique’s computing efficiency across several datasets, offering information on its effectiveness throughout both the training and testing stages. NSL-KDD, UNSW-NB15, and CIC-IDS2017 are among the datasets. It shows the CPU usage and memory utilization in percentage, and training duration for each dataset and task (training or testing). Notably, the DRX approach uses different resources across datasets, demonstrating its flexibility in varied contexts. This information is critical for understanding the computing requirements of the DRX approach, allowing researchers and practitioners to make educated judgments regarding its applicability to various datasets and workloads.

### 5.5. K-Fold Cross-Validation

K-fold cross-validation is a critical resampling approach in machine learning, primarily used to assess the performance and flexibility of prediction models. This strategy systematically splits a dataset into K equivalent subsets, referred to as “folds”, before training and testing the model in a cyclical process. Each iteration entails using K − 1 folds for model training and the remaining fold for testing, and the partitioning and evaluation cycle is repeated K times. The distinguishing feature is the full assessment over many subsets, which reduces the influence of data unpredictability inherent in a single train–test split and provides a more robust performance estimate. This approach assists in predicting how well a model will perform on new, previously unknown data. Beyond performance estimation, K-fold cross-validation is a useful technique for solving common model-building difficulties. It aids in the diagnosis of overfitting and underfitting problems by highlighting differences in performance across different training–test splits. Furthermore, while fine-tuning a model’s hyperparameters, the approach allows for the systematic investigation of various parameter configurations across numerous data subsets. This makes it easier to choose appropriate hyperparameter values, which improves the model’s generalization capabilities.

K-fold cross-validation provides maximum exploitation of every data point in cases where data availability is restricted, which is critical for tiny datasets. Furthermore, by injecting randomness through fold shuffling, it aids in minimizing bias in performance evaluation, particularly in datasets with intrinsic order or structure. In essence, K-fold cross-validation is a foundational method that improves model dependability. Using a 10-fold cross-validation technique instead of a lesser number of folds, such as 1, 2, 3, or 5, improves the model assessment resilience and reliability. By averaging over a greater number of evaluations, lowering fluctuation in performance measures, and providing a statistically significant sample size for analysis, the technique with 10 folds provides a more consistent estimate of performance. This not only smooths out any model performance fluctuation owing to random data splits but also provides a more nuanced understanding of the model’s behavior across varied dataset subsets. The 10-fold cross-validation approach is very useful for spotting complicated problems such as overfitting or underfitting and making more trustworthy conclusions about the model’s generalizability.

The K-fold cross-validation results for the supplied datasets are depicted in Table 12, that shows the model’s mean cross-validation scores and training accuracy. The model has a mean cross-validation score of 0.9772 in the NSL-KDD dataset, indicating a good level of consistency and performance over multiple folds. On this dataset, the training accuracy is given as 0.9976, reflecting the model’s success in learning patterns and features during the training phase. When applied to the UNSW-NB15 dataset, the model achieves an even better mean cross-validation score of 0.991, demonstrating its robustness and generalizability. On this dataset, the training accuracy is 0.99, confirming the model’s capacity to reliably categorize instances during the training phase. Similarly, the model earns a mean cross-validation score of 0.99 for the CIC-IDS2017 dataset, confirming its consistent performance across multiple subsets of the data. The training accuracy on this dataset is likewise 0.99, demonstrating the model’s capacity to reach high levels of accuracy throughout the training phase. Overall, these findings indicate that the proposed model works well on these datasets, with high training accuracy and great cross-validation scores.

## 6. Discussion

In this section, we conduct a comparative analysis of the DRX ensemble classifier with several other ensemble classifier techniques that have utilized the NSL-KDD, UNSW-NB15, or CIC-IDS2017 dataset for experimental purposes. Figure 6 and Figure 7 show the average of the performance metrics achieved by the proposed technique for the training dataset and testing dataset, respectively.

Upon examination, we found that the authors had proposed ensemble classifier techniques and tested using the NSL-KDD dataset. The techniques proposed by Lian et al. [39], Gao et al. [40], Alhowaide et al. [43], Kunar et al. [44], Otoum et al. [46], Das et al. [45], and Yousefnezhad et al. [33] resulted in 99.23%, 84.23%, 98.0%, 99.72%, 99.40%, 99.10%, and 99.80% accuracy, respectively. Remarkably, the DRX technique surpasses the performance of most other techniques, attaining an accuracy rate of 99.88%. This represents a substantial improvement in accuracy when compared to the majority of the techniques evaluated. Furthermore, the DRX technique exhibits a slightly higher accuracy rate of 0.08% compared to the best-performing approach presented in [33]. In terms of false positive rate, Otoum et al. [46] achieved the lowest FPR of 0.013, outperforming Das et al. [45] and Kunar et al. [44] with FPR values of 0.03 and 0.088, respectively. Notably, our proposed DRX solution achieves the lowest FPR of 0.003 among the ensemble techniques.

Various ensemble classifier techniques have been proposed in the last couple of years that use the UNSW-NB15 dataset to test their proposed scheme. The ensemble classifier technique proposed in [53] achieved a higher accuracy compared to schemes proposed in [52,54]. The higher accuracy achieved was 99.32%. Our proposed ensemble classifier outperformed and achieves the highest accuracy, with an average of 99.93% for the classes available in the UNSW-NB15 dataset. There are a few techniques that were tested against the FPR. Some of them received a high FPR, such as the scheme proposed in [35,51], which obtained a 9.23% and 11.3% false positive rate, respectively. The technique proposed in [52] achieved a lower FPR of 0.017%. Our proposed scheme outperformed these solutions and achieves the lowest FPR, which is 0.001 for the UNSW-NB15 dataset.

Similarly, authors proposed classification techniques that were tested using the CIC-IDS2017 dataset. The proposed ensemble classifier technique of [47] achieved 86.5% accuracy, while model presented in [66] achieved 99.86% accuracy. Other techniques that were proposed in [25,34,48,49], received, 97.72%, 99.89%, 99.95%, and 98.62%accuracy, while our proposed scheme is 99.98% accurate during the classification. The reported FPR using the CIC-IDS2017 dataset is 0.12 and 0.013, acquired by techniques proposed in [25,34], which is further reduced to 0.00012 by our proposed ensemble classifier.

Overall, the DRX-based voting ensemble classifier showcases superior performance with its significantly higher accuracy rate as shown in Figure 6 and Figure 7 and the lowest FPR when compared to other ensemble classifier techniques evaluated in this study.

The suggested DRX ensemble classifier takes an innovative approach to improving the security and efficiency of metaverse systems. The necessity for resilient and adaptable security methods is critical in the metaverse, where virtual and augmented reality intersect with the physical world. The DRX solution demonstrates its ability to reinforce the metaverse against cyber threats and attacks by reaching a phenomenal accuracy rate of 99.88%. Its excellent performance assures virtual environment integrity, protecting users, assets, and interactions inside the metaverse. Furthermore, in the metaverse, where several data streams and interactions occur at the same time, the ensemble aspect of the DRX approach is helpful. Its capacity to combine and analyze data from several sources offers a thorough awareness of the metaverse’s dynamic landscape, which aids in threat detection and anomaly identification. This novel feature distinguishes the DRX ensemble classifier as an important asset for security frameworks in the growing metaverse ecology. The DRX ensemble classifier’s integration extends its benefits to fog and edge computing environments, where processing and decision making take place closer to the data source. For real-time applications in fog and edge computing, decreasing latency is critical. The rapid and precise decision-making mechanism of the DRX ensemble classifier is well suited for these situations, providing timely reactions to security risks or abnormalities. This is especially important in situations requiring a fast response, such as critical infrastructure or IoT deployments. Bandwidth restrictions are common in edge computing scenarios. The DRX system minimizes bandwidth use by utilizing ensemble methods. Instead of sending enormous amounts of raw data to a central server, the ensemble classifier examines data locally and sends only relevant insights. This not only saves bandwidth but also improves the system’s overall efficiency. Edge computing settings are dynamic, with devices continually joining and departing the network. The versatility of the DRX ensemble classifier enables consistent performance even while the edge architecture develops. Its capacity to adapt to changes in network topology dynamically makes it a durable choice for edge computing installations. Given the dispersed nature of the infrastructure, security is a significant problem in edge computing. The DRX ensemble classifier’s extensive security features help to protect edge devices from a variety of cyber attacks. Its high accuracy rate reduces false positives and negatives, ensuring the accurate detection of security occurrences while not taxing edge resources.

Table 11 shows the computational resources of the DRX technique. The presented data underscore the computational characteristics of the DRX model. For instance, during the training phase on the NSL-KDD dataset, the CPU utilization was 30.4%, with a minimal memory utilization of 0.2% and a training time of 21.98 s. Similarly, the testing phase on the same dataset exhibited a CPU utilization of 30.0%, minimal memory usage (0.2%), and a short testing time of 1.47 s. For the UNSW-NB15 dataset, the training phase demonstrated a slightly higher CPU utilization at 37%, still with a manageable memory utilization of 0.7% and a training time of 43.5 s. The testing phase on this dataset maintained reasonable CPU and memory utilization percentages (35% and 0.7%, respectively) and a testing time of 2.05 s. Lastly, the CIC-IDS2017 dataset showcased a higher CPU utilization during training (39%) but with a commendably low memory utilization of 0.5% and a training time of 55 s. The testing phase on the same dataset reported a CPU utilization of 37%, minimal memory usage (0.4%), and a short testing time of 1.21 s. By discussing these results, we highlight the model’s computational efficiency and suitability for deployment in resource-constrained IoT devices.

The proposed model performs effectively against different collected datasets in different IoT environments. One can test the possibility of a generalized solution based on DRX in the future, where the proposed method will be evaluated in a more dense and realistic environment to check the generalization of the model.

Beyond the datasets employed in the assessment, the suggested DRX-based methodology shows significant promise for generalizability across varied and dynamic real-world IoT scenarios. The persistent high accuracy rates achieved across numerous datasets, including NSL-KDD, UNSW-NB15, and CIC-IDS2017 (99.88%, 99.93%, and 99.98% accuracy, respectively), highlight the DRX ensemble classifier’s resilient and generalized performance in diverse intrusion detection settings. The model’s ability to handle false positive rates (FPRs) well, with astonishingly low values of 0.003, 0.001, and 0.00012 for NSL-KDD, UNSW-NB15, and CIC-IDS2017, respectively, demonstrates its potential to decrease false alarms—a critical component for real-world implementation. The computational efficiency data given demonstrate moderate CPU and memory use percentages throughout training and testing stages, demonstrating suitability for resource-constrained IoT applications. Furthermore, DRX’s incorporation into metaverse systems, fog computing, and edge computing environments illustrates its flexibility in various and dynamic situations, assuring optimal performance in real-time decision making, decreased latency, and efficient bandwidth consumption. The robust security features of the model help to secure edge devices in security-sensitive edge computing situations. Overall, the DRX-based approach provides a viable and adaptable option for intrusion detection in a wide range of real-world IoT contexts.

## 7. Conclusions

This research paper introduces a novel machine-learning-based ensemble voting classification technique known as DRX (decision tree, random forest, and XGBoost) for efficiently handling multi-class datasets. The proposed methodology is evaluated using widely used datasets, including NSL-KDD, UNSW-NB15, and CIC-IDS2017, with performance assessed using various evaluation metrics, such as accuracy, precision, recall, F1-score, and false positive rate. The experimental results clearly demonstrate the effectiveness of the DRX technique, achieving remarkable accuracy rates of 99.88%, 99.93%, and 99.98% on the NSL-KDD, UNSW-NB15, and CIC-IDS2017 datasets, respectively, surpassing the performance of other ensemble classifier approaches. Additionally, the analysis of false positive rates reveals that the DRX technique achieves the lowest rate when compared to other ensemble classifier methods on these datasets.

While this work primarily focuses on testing the DRX ensemble classifier on well-established multi-class datasets, future research avenues can explore its application on datasets with a higher number of classes. Furthermore, investigating the time utilization of the DRX technique in comparison to other classifiers would provide valuable insights. Researchers can also explore alternative combinations of machine learning techniques involving two or more algorithms. Moreover, the proposal of hybrid ensemble classifier techniques that integrate both machine learning and deep learning algorithms could be further explored. Such approaches could be evaluated on datasets suitable for machine learning techniques, as well as those derived from IoT environments.

Overall, the DRX ensemble classifier technique shows promising results and holds potential as an effective tool for network intrusion detection systems (NIDSs), particularly in the evolving landscape of IoT security. The research not only contributes to the field of cybersecurity but also underscores the importance of continually advancing intrusion detection techniques to address the security challenges brought about by the ever-expanding IoT ecosystem.

## Figures and Tables

**Figure 1 sensors-24-00127-f001:**
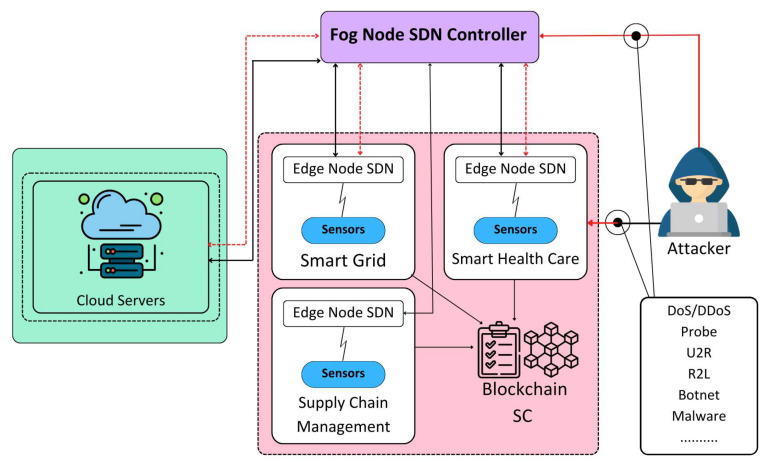
Security requirements for the IoT-enabled cloud platform.

**Figure 2 sensors-24-00127-f002:**
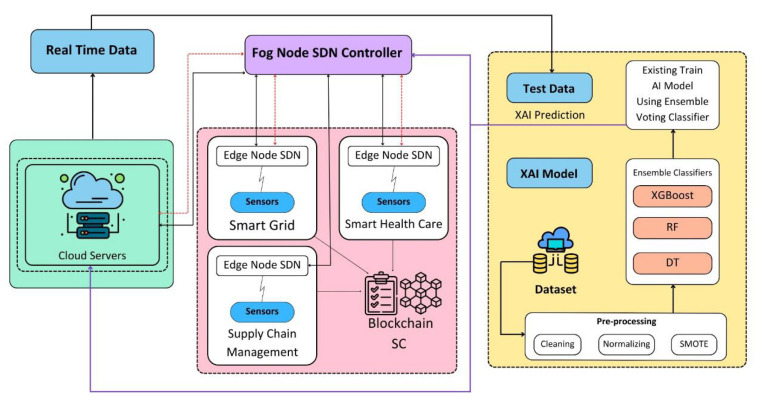
Architecture diagram of NIDS in IoT-enabled cloud platform.

**Figure 3 sensors-24-00127-f003:**
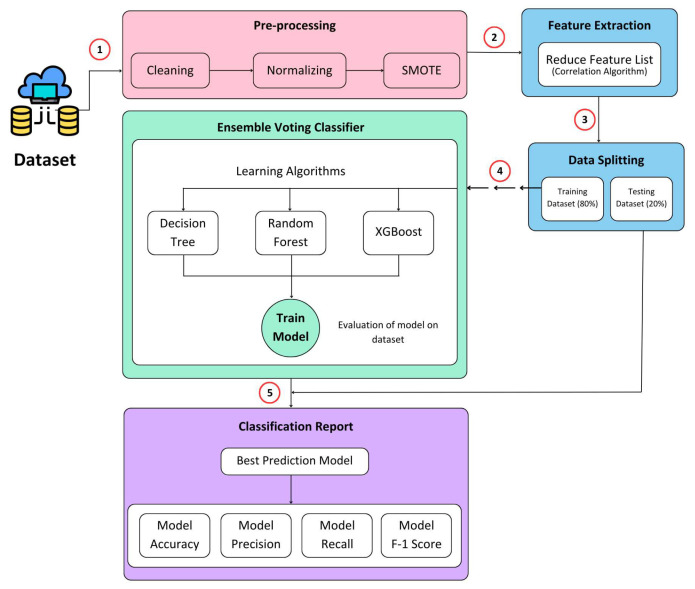
Proposed ML-based NIDS technique using ensemble voting classifier.

**Figure 4 sensors-24-00127-f004:**
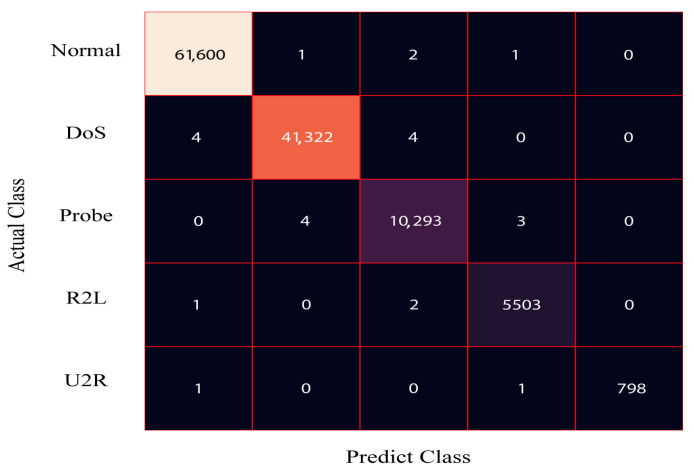
Confusion matrix for the train data of the NSL-KDD dataset.

**Figure 5 sensors-24-00127-f005:**
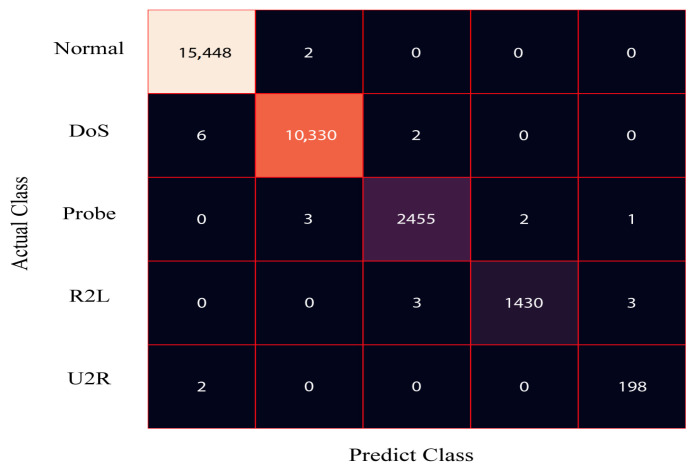
Confusion matrix for the test data of the NSL-KDD dataset.

**Figure 6 sensors-24-00127-f006:**
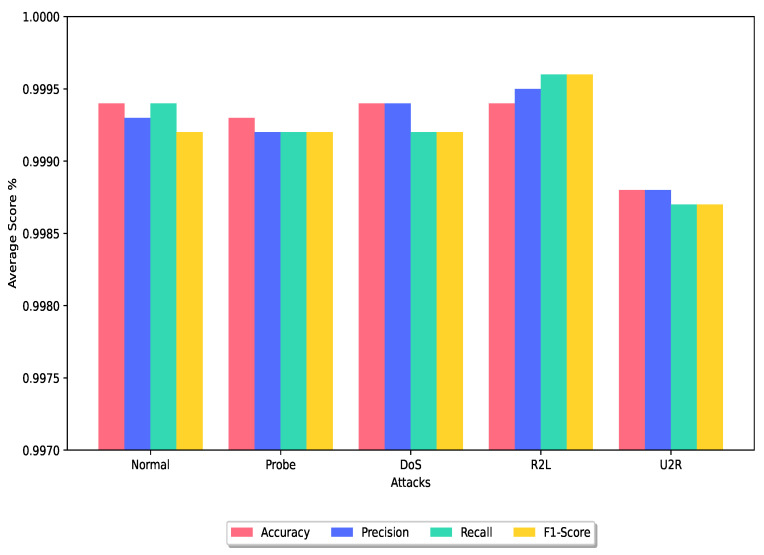
Average analysis of the DRX technique on the train dataset.

**Figure 7 sensors-24-00127-f007:**
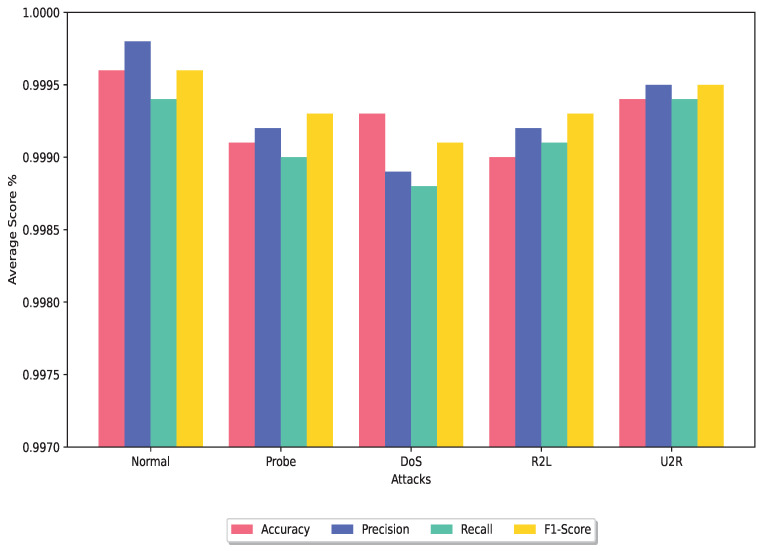
Average analysis of the DRX classifier using test dataset.

**Table 1 sensors-24-00127-t001:** Overview of the ensemble-classifier-based NIDS techniques proposed for securing IoT-enabled environment.

Author	Dataset	Technique	Ensemble Classifier	Evaluation Metrics	Achieved
Lian et al. [39]	KDD-CUP99 NSL-KDD	DT recursive feature elimination	DT, RF, AdaBoost	Precision, Acc, F1-score	Avg. 99.22% Accuracy
Gao et al. [40]	NSL-KDD	Adaptive ensemble ML model	MultiTree algorithm	Precision, Acc, Rec, F1-score	84.23% Accuracy
Alhowaide et al. [43]	NSL-KDD UNSW-NB15 BoTNeTIoT BoTIoT	Automatic model selection method	SVM, DT, …	Accuracy, F1-score	Avg. 98.25% Accuracy
Yousefnezhad et al. [33]	UNSW-NB15 CICIDS2017 NSL-KDD	Dempster– Shafer method	KNN, SVM	Precision, Acc, F1-score, Rec	Avg. 99.80% Accuracy
Kunal et al. [44]	NSL-KDD	WEKA knowledge analysis	KNN, RT, RF	Accuracy, Recall, FPR	99.68% Accuracy, 0.003 FPR
Das et al. [45]	NSL-KDD	WEKA majority voting	SVM, KNN, NN, C4.5 or DT	Precision, Acc, Rec, FPR	99.10% Accuracy, 0.088 FPR
Otoum et al. [46]	NSL-KDD	Dependent BCC-based IDS	RF, DBSCAN, RBM	Accuracy, FPR	99.40% Accuracy, 0.87 FPR
Chen et al. [47]	MAWILab CIC-IDS2017	Similarity-aware IDS	ADSIM	Precision, Acc, Rec, F1-score	86.50% Accuracy
Karna et al. [48]	NSL-KDD CIC-IDS2017	Filter-based selection	DT, RF, ET	Precision, Acc, F1-score	Avg. 99.51% Accuracy
Seth et al. [49]	CIC-IDS2017	Hybrid feature selection	Gradient boosting machine	Precision, Acc, Rec, F1-score	97.72% Accuracy
Zhou et al. [25]	NSL-KDD, AWID, CIC-IDS2017	CFS-BA-ensemble method	C4.5 or DT, RF, FPA	Precision, Acc, F1-score, FPR	Avg. 99.80% Accuracy, Min 0.12 FPR
Zhang et al. [34]	KDDCup-99 NSL-KDD UNSW-NB15 CIC-IDS2017	Feature fusion and stacking mechanism	DT, RF	Precision, Acc, Rec, F1-score FPR	Avg. 91.40% Accuracy, Min 0.013 FPR
Kabir et al. [50]	UNSW-NB15	Mutual information gain selection	Extra tree (ET) classifier	Accuracy	96.24% Accuracy
G. Kaur [51]	NSL-KDD UNSW-NB15	Weighted-voting-based ensemble	RF, AdaBoost	F1-score, Acc, FPR	Avg. 89.50% Accuracy, 9.23 FPR
Maniriho et al. [52]	NSL-KDD, UNSW-NB15	Gain ratio feature Evaluator	Random committee	Precision, Acc, FPR	Avg. 98.80% Accuracy, 0.033 FPR
Gangula et al. [53]	NSL-KDD, UNSW-NB15	Enhanced flower pollination algorithm	DT, RF, SVM	Accuracy	Avg. 99.49% Accuracy
Tama et al. [35]	NSL-KDD UNSW-NB15 CSIC-2010v2 CIC-IDS2017	Stacked ensemble classifier	RF, GBM, XGBoost	Acc, FPR	Avg. 95.85% Accuracy, Min 0.46 FPR

“SVM”: support vector machine, “RT”: random tree, “KNN”: k-nearest neighbor, “DNN”: deep neural network,“RBM” stands for restricted Boltzmann machine, “RF” refers to random forest, “DT” represents decision tree, “LR”represents logistic regression, “DBSCAN” stands for density-based spatial clustering of applications with noise,“RFE” denotes recursive feature elimination, “DELM” refers to deep extreme learning machine, “GBM”: gradientboosting machine, “FPA”: forest by penalizing attributes, “BCC”: Bayesian combination classification.

**Table 2 sensors-24-00127-t002:** Training and testing dataset after pre-processing the NSL-KDD dataset.

Attack Type	Training Dataset (80%)	Test Dataset (20%)
Normal	61,604	15,450
Probe	10,300	2462
DoS	41,330	10,338
U2R	800	200
R2L	5506	1436

**Table 3 sensors-24-00127-t003:** Result analysis for DRX algorithm for 80% training on NSL-KDD dataset.

Label	Accuracy	Precision	Recall	F1-Score
Normal	0.9994	0.9993	0.9994	0.9992
Probe	0.9993	0.9992	0.9992	0.9992
DoS	0.9994	0.9994	0.9992	0.9992
R2L	0.9994	0.9995	0.9996	0.9996
U2R	0.9988	0.9988	0.9987	0.9987

**Table 4 sensors-24-00127-t004:** Results for DRX algorithm on 20% test data.

Label	Accuracy	Precision	Recall	F1-Score
Normal	0.9996	0.9998	0.9994	0.9996
Probe	0.9991	0.9992	0.9990	0.9993
DoS	0.9993	0.9989	0.9988	0.9991
R2L	0.9990	0.9992	0.9991	0.9993
U2R	0.9994	0.9995	0.9994	0.9995

**Table 5 sensors-24-00127-t005:** Training and testing dataset after pre-processing the UNSW-NB15 dataset.

Attack Type	Training Dataset (80%)	Test Dataset (20%)
Normal	74,387	18,613
Generic	47,334	11,537
Exploits	35,445	9080
Fuzzers	19,415	4831
Dos	13,061	3292
Reconnaissance	11,135	2852
Analysis	2158	519
Backdoor	1861	468
Shellcode	1202	309

**Table 6 sensors-24-00127-t006:** Result analysis for DRX algorithm for 80% training on UNSW-NB15 dataset.

Label	Accuracy	Precision	Recall	F1-Score
Normal	1.0	1.0	1.0	1.0
Generic	1.0	1.0	1.0	1.0
Exploits	1.0	1.0	1.0	1.0
Fuzzers	1.0	1.0	1.0	1.0
Dos	1.0	1.0	1.0	1.0
Reconnaissance	1.0	1.0	1.0	1.0
Analysis	1.0	1.0	1.0	1.0
Backdoor	0.9999	0.9998	0.9999	0.9997
Shellcode	0.9999	0.9999	0.9999	0.9999

**Table 7 sensors-24-00127-t007:** Result analysis for DRX algorithm for 20% test on UNSW-NB15 dataset.

Label	Accuracy	Precision	Recall	F1-Score
Normal	1.0	1.0	1.0	1.0
Generic	1.0	1.0	1.0	1.0
Exploits	1.0	1.0	1.0	1.0
Fuzzers	1.0	1.0	1.0	1.0
Dos	1.0	1.0	1.0	1.0
Reconnaissance	1.0	1.0	1.0	1.0
Analysis	0.9999	0.9998	0.9999	0.9997
Backdoor	0.9998	0.9998	0.9999	0.9998
Shellcode	0.9999	0.9997	0.9998	0.9999

**Table 8 sensors-24-00127-t008:** Training and testing dataset after pre-processing the CIC-IDS 2017 dataset.

Attack Type	Training Dataset	Test Dataset
Normal	91,607	22,902
DoS/DDoS	49,386	12,346
PortScan	18,491	4623
Brute Force	6223	1556
Web Attack	1558	389
Botnet ARES	1694	424
Infiltration	29	7

**Table 9 sensors-24-00127-t009:** Result analysis for DRX algorithm for 80% train on CIC-IDS2017 dataset.

Label	Accuracy	Precision	Recall	F1-Score
Normal	1.0	1.0	1.0	1.0
Web Attack	1.0	1.0	1.0	1.0
Brute Force	1.0	1.0	1.0	1.0
DoS/DDoS	99.99	99.99	99.99	99.99
Infiltration	1.0	1.0	1.0	1.0
PortScan	1.0	1.0	1.0	1.0
Botnet ARES	99.99	99.99	99.99	99.99

**Table 10 sensors-24-00127-t010:** Result analysis for DRX algorithm for 20% test on CIC-IDS2017 dataset.

Label	Accuracy	Precision	Recall	F1-Score
Normal	1.0	1.0	1.0	1.0
Web Attack	1.0	1.0	1.0	1.0
Brute Force	1.0	1.0	1.0	1.0
DoS/DDoS	99.99	99.99	99.99	99.99
Infiltration	1.0	1.0	1.0	1.0
PortScan	99.99	99.99	99.99	99.99
Botnet ARES	99.99	99.99	99.99	99.99

**Table 11 sensors-24-00127-t011:** Computational efficiency of DRX technique.

Dataset	Task	CPU Utilization%	Memory Utilization%	Training Time (s)
NSL-KDD	Training	30.4	0.2	21.98
NSL-KDD	Testing	30.0	0.2	1.47
UNSW-NB15	Training	37	0.7	43.5
UNSW-NB15	Testing	35	0.7	2.05
CIC-IDS2017	Training	39	0.5	55
CIC-IDS 2017	Testing	37	0.4	1.21

**Table 12 sensors-24-00127-t012:** K-fold cross-validation.

Dataset	Mean Cross-Validation Score	Training Accuracy
NSL-KDD	0.9772	0.9976
UNSW-NB15	0.991	0.995
CIC-IDS2017	0.99	0.99

## Data Availability

The datasets used in this research are publicly available and no new dataset is generated in this research.

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
