# Peer review of "Enhancing Network Intrusion Detection Using an Ensemble Voting Classifier for Internet of Things"

_sensors, 2023, doi:10.3390/s24010127_

Round 1

Reviewer 1 Report

Comments and Suggestions for Authors

The purpose of the paper is commendable. However, the references appear to be hastily composed and should be cited appropriately within the paper. It is essential to include the latest references and ensure accuracy in citation format. Additionally, please exercise caution in reviewing the Keywords section, where an extra ')' symbol is present. Furthermore, many journal references in the bibliography lack the indication of the issue number (No) and only specify the volume (Vol.). It is recommended to rectify this by including the issue number for comprehensive referencing.

Author Response

Reviewer 1: The purpose of the paper is commendable. However, the references appear to be hastily composed and should be cited appropriately within the paper. It is essential to include the latest references and ensure accuracy in citation format. Additionally, please exercise caution in reviewing the Keywords section, where an extra ')' symbol is present. Furthermore, many journal references in the bibliography lack the indication of the issue number (No) and only specify the volume (Vol.). It is recommended to rectify this by including the issue number for comprehensive referencing.

Reviewer#1, Concern # 1 The purpose of the paper is commendable..

 Author response:  Thank you and appricaited.

Author Action:  Not Required

Reviewer#1, Concern # 2 However, the references appear to be hastily composed and should be cited appropriately within the paper.. It is essential to include the latest references and ensure accuracy in citation format.

 Author response:  Thank you for your suggestion.

Author Action:  we have updated all the references as suggested . We have improved the references. New research articles are also added as per the requirement of the improved version. We had proofread the document to ensure accuracy in the citation format.

Reviewer#1, Concern # 3 Additionally, please exercise caution in reviewing the Keywords section, where an extra ')' symbol is present.

 Author response:  Thank you again for correting us.

Author Action:  We have updated and corrected .We had proofread the document to ensure that there are no extra punctuation marks. Further, extra ‘)’ symbol is removed in keyword section.

Reviewer#1, Concern # 4 Furthermore, many journal references in the bibliography lack the indication of the issue number (No) and only specify the volume (Vol.). It is recommended to rectify this by including the issue number for comprehensive referencing.

 Author response:  Thank you again for your suggestion on references.

Author Action:  : We had added the issue no in the bibliography file in overleaf but unfortunately MDPI reference style did not include issue number. We went through the MDPI manual available at https://www.mdpi.com/authors/layout. According to this, “the MDPI reference style omits the issue number”.

Reviewer 2 Report

Comments and Suggestions for Authors

To counter various types of IoT cyberattacks, detecting anomalies in the network is critical to protecting applications. Based on the application of voting classifiers in machine learning, the authors combine the advantages of decision trees, random forests, and XGBoost algorithms to propose a novel classification technique based on the DRX method, which improves the accuracy and precision of network intrusion detection systems. Finally, the authors conducted experiments using the NSL-KDD, UNSW- NB15, and CIC-IDS2017 datasets. Experimental results show that the algorithm has significant results. However, we believe that this paper has some shortcomings in some aspects:

1、We believe that the authors have described the innovativeness of this paper only from the experimental part. We suggest that the authors describe the innovation comprehensively in the revised manuscript.

2、The references cited in this paper are excessive and could have been cut in moderation.

3、Authors should use latex format for the description of the algorithm.

Comments on the Quality of English Language

None.

Author Response

Reviewer 2: 

To counter various types of IoT cyberattacks, detecting anomalies in the network is critical to protecting applications. Based on the application of voting classifiers in machine learning, the authors combine the advantages of decision trees, random forests, and XGBoost algorithms to propose a novel classification technique based on the DRX method, which improves the accuracy and precision of network intrusion detection systems. Finally, the authors conducted experiments using the NSL-KDD, UNSW- NB15, and CIC-IDS2017 datasets. Experimental results show that the algorithm has significant results. However, we believe that this paper has some shortcomings in some aspects:

1、We believe that the authors have described the innovativeness of this paper only from the experimental part. We suggest that the authors describe the innovation comprehensively in the revised manuscript.

2、The references cited in this paper are excessive and could have been cut in moderation.

3、Authors should use latex format for the description of the algorithm.

Reviewer#2, Concern # 1:  We believe that the authors have described the innovativeness of this paper only from the experimental part. We suggest that the authors describe the innovation comprehensively in the revised manuscript.

Author response: Thank you and we are greatful for your kind review and comments.

Author Action:  We have now added a detail description of innovation in a paragraph is presented on Page no 19 to 22.

Reviewer#2, Concern # 2:  The references cited in this paper are excessive and could have been cut in moderation.

Author response: Thank you again for your suggestion.

Author Action:  We had proofread the document to lessen the references. The count is changed from 84 to 67 in the modified manuscript.

Reviewer#2, Concern # 3:  Authors should use latex format for the description of the algorithm.

Author response: Thank you again for your suggestion for description of the algorithm.

Author Action:  We had used latex format for the description of the algorithm on page 13.

Reviewer 3 Report

Comments and Suggestions for Authors

Based on the contents of the manuscript "Enhancing Network Intrusion Detection using an Ensemble Voting Classifier for Internet of Things," here are recommendations for major revision:

1. Detailed Comparison with Existing Methods

- Literature Review Gaps: The literature review (Section 2) needs to be expanded to include a broader and more detailed comparison with existing methods. For instance, while the authors mention several techniques (like ML-based NIDS using RepTree, single classifier technique using XGBoost, etc.), a deeper analysis on how these methods compare to the DRX-based approach in terms of effectiveness, efficiency, and applicability in IoT is lacking.

- Comparison of metrics within Table 1 on different studies are not meaningful when different datasets are used. Maybe group studies by datasets and compare again?

- Specific Comparisons Required: The manuscript should specifically address how the DRX approach improves upon these existing methods. Are there benefits in terms of accuracy, computational efficiency, or adaptability to IoT environments? This needs to be explicitly stated and backed by data.

2. Deeper Analysis of Methodology

- Dataset Justification: While the use of NSL-KDD, UNSW-NB15, and CIC-IDS2017 datasets appears appropriate, the manuscript doesn't thoroughly explain the selection criteria (Section 3.1). The authors should justify how these datasets comprehensively represent intrusion scenarios in IoT environments. 

- Computational Resource Impact: There's a notable absence of discussion on the computational resources required for the DRX model, particularly in terms of real-time application in IoT environments (Section 4.1). The authors should assess the model's computational efficiency and suitability for deployment in resource-constrained IoT devices.

3. Enhanced Critical Analysis of Results

- Overfitting Concerns: The authors' extremely high accuracy rates reported raise questions about overfitting. This issue needs to be critically analyzed and addressed (Section 5). The manuscript should include strategies employed to prevent overfitting and validate the model's generalizability.

- Real-world Applicability: The discussion section (Section 6) should include a critical evaluation of the model's applicability in real IoT environments. This includes assessing the model's performance in dynamic, heterogeneous IoT networks and its scalability.

4. Improved Clarity in Writing

- Technical Terms and Jargon: The manuscript is dense with technical jargon, which could be made more accessible with simpler language and better explanations of complex terms (throughout the manuscript).

- Structural Coherence: Some sections would benefit from clearer organization and more logical flow of information. This includes a more coherent transition between the methodology, results, and discussion sections.

- The equations in section 4.2 about accuracy, false positive etc are not necessary, because they are not the author's original definitions or equations.

Author Response

Reviewer 3

Reviewer#3, Concern # 1:  Literature Review Gaps: The literature review (Section 2) needs to be expanded to include a broader and more detailed comparison with existing methods. For instance, while the authors mention several techniques (like ML-based NIDS using RepTree, single classifier technique using XGBoost, etc.), a deeper analysis on how these methods compare to the DRX-based approach in terms of effectiveness, efficiency, and applicability in IoT is lacking.

Author response: Thank you and updated accordingly.

Author Action:  We updated the manuscript by improving the writting style of the article as suggested. Detailed comparision is provided in the section Discussion  on page no 21 and 22.

Reviewer#3, Concern # 2:  Comparison of metrics within Table 1 on different studies are not meaningful when different datasets are used. Maybe group studies by datasets and compare again?

Author response: Thank you and updated accordingly.

Author Action:  Table 1 provides the overview of the techniques already grouped studied by datasets.The detail comparison of the proposed method against different datasets are discussed in section 5.

Comparison on NSL-KDD with different techniques is presented on page no 19.

Comparison on UNSW-NB15 with differrnt technquies is presented on page no 21.

Comparison on CIC-IDS 2017 with different technquies is presented on page no 21.

Reviewer#3, Concern # 3:  Specific Comparisons Required: The manuscript should specifically address how the DRX approach improves upon these existing methods. Are there benefits in terms of accuracy, computational efficiency, or adaptability to IoT environments? This needs to be explicitly stated and backed by data.

Author response: Thank you for your concern.

Author Action:  We discussed the specific comparisons in the section 6.They are explained in terms of accuracy, computational efficiency, or adaptability to IoT environments. The discussion based on the results acquired from experimental analysis.

Reviewer#3, Concern # 4:  Dataset Justification: While the use of NSL-KDD, UNSW-NB15, and CIC-IDS2017 datasets appears appropriate, the manuscript doesn't thoroughly explain the selection criteria (Section 3.1). The authors should justify how these datasets comprehensively represent intrusion scenarios in IoT environments. 

Author response: Thank you for your concern about Dataset justification.

Author Action:  We updated and justified the datasets requirement of NSL-KDD, UNSW-NB15, and CIC-IDS2017 datasets intrusion scenarios in IoT environment on page 8 and 9.

Reviewer#3, Concern # 5: Computational Resource Impact: There's a notable absence of discussion on the computational resources required for the DRX model, particularly in terms of real-time application in IoT environments (Section 4.1). The authors should assess the model's computational efficiency and suitability for deployment in resource-constrained IoT devices.

Author response: Thank you.

Author Action:  We discussed the computational resource calculation in section 4.1.

Reviewer#3, Concern # 6: Overfitting Concerns: The authors' extremely high accuracy rates reported raise questions about overfitting. This issue needs to be critically analyzed and addressed (Section 5). The manuscript should include strategies employed to prevent overfitting and validate the model's generalizability.

Author response: Thank you

Author Action: We updated and discussed the overfitting concerns in section 5.We have performed K-Fold cross validation.It proved that the proposed strategy is not overfit or underfit.

Reviewer#3, Concern # 8: Technical Terms and Jargon: The manuscript is dense with technical jargon, which could be made more accessible with simpler language and better explanations of complex terms (throughout the manuscript).

Author response: Thank you .

Author Action:  We revised manuscript and proofread as well.

Reviewer#3, Concern # 9: Structural Coherence: Some sections would benefit from clearer organization and more logical flow of information. This includes a more coherent transition between the methodology, results, and discussion sections.

Author response: Thank you for your concern.

Author Action: Our paper is represented in the formation is provided i.e firstly methodology is explained, secondly results are placed  and finally discussion is provided.

Reviewer#3, Concern # 10: The equations in section 4.2 about accuracy, false positive etc are not necessary, because they are not the author's original definitions or equations.

Author response: Thank you

Author Action:  Many researchers use the same equations in their research paper as a performance metrics.We have also mentioned the same in our research article.

Reviewer 4 Report

Comments and Suggestions for Authors

Generalization of the Model: The paper presents impressive results using the NSL-KDD, UNSW-NB15, and CIC-IDS2017 datasets. However, can you elaborate on the generalizability of the proposed DRX-based technique in diverse and dynamic real-world IoT environments beyond these datasets?

Comparison with Other Techniques: Your study benchmarks the DRX approach against other ensemble classifiers. Could you discuss how the DRX approach compares to advanced deep learning techniques, particularly in handling unstructured big data in IoT networks?

Scalability and Real-Time Application: Considering the computational complexity of the DRX approach, how scalable is this model in real-time IoT environments, especially when dealing with large-scale networks and high-frequency data streams?

Comments on the Quality of English Language

The Quality of English Language is good

Author Response

Reviewer 4

Reviewer#4, Concern # 1:  Generalization of the Model: The paper presents impressive results using the NSL-KDD, UNSW-NB15, and CIC-IDS2017 datasets. However, can you elaborate on the generalizability of the proposed DRX-based technique in diverse and dynamic real-world IoT environments beyond these datasets?

Author response: Thank you and will update accordingly.

Author Action:  We have discussed the generalization of model in section 6 discuss on page no 22.

Reviewer#4, Concern # 2:  Comparison with Other Techniques: Your study benchmarks the DRX approach against other ensemble classifiers. Could you discuss how the DRX approach compares to advanced deep learning techniques, particularly in handling unstructured big data in IoT networks?

Author response: Thank you and will update accordingly.

Author Action:  We compared our technique with various ensemble classifier techniques and detail is provided in section 6 on page no 22.

Reviewer#4, Concern # 3:  Scalability and Real-Time Application: Considering the computational complexity of the DRX approach, how scalable is this model in real-time IoT environments, especially when dealing with large-scale networks and high-frequency data streams?

Author response: Thank you and will update accordingly.

Author Action:  We discussed it in discussion section on page no 22..

Round 2

Reviewer 3 Report

Comments and Suggestions for Authors

Full acceptance recommended after major revision.